# CD146+ Endometrial-Derived Mesenchymal Stem/Stromal Cell Subpopulation Possesses Exosomal Secretomes with Strong Immunomodulatory miRNA Attributes

**DOI:** 10.3390/cells11244002

**Published:** 2022-12-10

**Authors:** Clarissa Leñero, Lee D. Kaplan, Thomas M. Best, Dimitrios Kouroupis

**Affiliations:** 1Department of Orthopaedics, UHealth Sports Medicine Institute, University of Miami Miller School of Medicine, Miami, FL 33146, USA; 2Diabetes Research Institute & Cell Transplant Center, University of Miami Miller School of Medicine, Miami, FL 33136, USA

**Keywords:** mesenchymal stem/stromal cells (MSCs), CD146 immunoselection, endometrium, exosomes, miRNAs, immunomodulation, inflammation, macrophage polarization, peripheral blood mononuclear cell functionality

## Abstract

The perivascular localization of endometrial mesenchymal stem/stromal cells (eMSC) allows them to sense local and distant tissue damage, promoting tissue repair and healing. Our hypothesis is that eMSC therapeutic effects are largely exerted via their exosomal secretome (eMSC EXOs) by targeting the immune system and angiogenic modulation. For this purpose, EXOs isolated from Crude and CD146+ eMSC populations were compared for their miRNA therapeutic signatures and immunomodulatory functionality under inflammatory conditions. eMSC EXOs profiling revealed 121 in Crude and 88 in CD146+ miRNAs, with 82 commonly present in both populations. Reactome and KEGG analysis of miRNAs highly present in eMSC EXOs indicated their involvement among others in immune system regulation. From the commonly present miRNAs, four miRNAs (hsa-miR-320e, hsa-miR-182-3p, hsa-miR-378g, hsa-let-7e-5p) were more enriched in CD146+ eMSC EXOs. These miRNAs are involved in macrophage polarization, T cell activation, and regulation of inflammatory cytokine transcription (i.e., TNF-α, IL-1β, and IL-6). Functionally, stimulated macrophages exposed to eMSC EXOs demonstrated a switch towards an alternate M2 status and reduced phagocytic capacity compared to stimulated alone. However, eMSC EXOs did not suppress stimulated human peripheral blood mononuclear cell proliferation, but significantly reduced secretion of 13 pro-inflammatory molecules compared to stimulated alone. In parallel, two anti-inflammatory proteins, IL-10 and IL-13, showed higher secretion, especially upon CD146+ eMSC EXO exposure. Our study suggests that eMSC, and even more, the CD146+ subpopulation, possess exosomal secretomes with strong immunomodulatory miRNA attributes. The resulting evidence could serve as a foundation for eMSC EXO-based therapeutics for the resolution of detrimental aspects of tissue inflammation.

## 1. Introduction

The functional layer of the human endometrium is a highly regenerative tissue undergoing monthly cycles of growth and differentiation, and it is maintained by perivascular endometrial-derived mesenchymal stem/stromal cells (eMSCs) [1,2,3]. In general, mesenchymal stem/stromal cells (MSCs) are non-hematopoietic cells showing ease of isolation, extensive proliferation capacity, and multilineage differentiation potential in vitro [4,5]. CD146 expression in MSCs plays a key role in the perivascular niche, skeletogenesis, and hematopoietic support in vivo [6,7,8]. On this basis, our previous data showed that the CD146 signature is correlated with innately higher MSC immunomodulatory and secretory capacities, and, thus, therapeutic potency in vivo [9]. Interestingly, among the first markers used to identify eMSCs possessing higher clonogenic capacity in vitro was CD146, strongly indicating their perivascular origin in intact full-thickness endometrium [10]. Specifically, SUSD2, a single marker distinguishing perivascular eMSCs from the surrounding endometrial stromal cells, has been found to co-express CD146 and CD140b, MSC/pericytic markers which play an important role in the cyclical regeneration of this highly regenerative tissue [1,11,12]. Furthermore, we and others have demonstrated that MSCs acquiring a potent immunomodulatory phenotype actively reverse both inflammation and fibrosis linked with macrophage polarization from an M1 in disease to an M2 phenotype [13,14,15,16,17,18].

Similarly to “medicinal signaling” activities of MSCs derived from other tissue sources [4,19], eMSCs’ perivascular localization allows them to sense local and distant tissue damage, migrate to affected sites, and promote endometrium repair and healing. In general, MSCs’ paracrine immunomodulatory and trophic (i.e., angiogenic, anti-fibrotic, anti-apoptotic, analgesic, and mitogenic) actions in vivo are exerted via secretion of abundant amounts of growth factors, cytokines, and extracellular vesicles [9,17,20,21,22]. Specifically, exosomes are a heterogeneous group of nano-sized (50–200 nm) extracellular vesicles (EVs), naturally secreted by cells that are immunologically inert. They contain many cell constituents, including DNA, RNA, lipids, metabolites, and cytosolic and cell-surface proteins [23,24]. MSC-derived exosomes mediate various biological functions attributed to MSC, such as tissue regeneration, intercellular communication, modulation of immunity, and cell signaling [25]. We and others have isolated and characterized exosomes from various MSC sources (i.e., infrapatellar fat pad, bone marrow, umbilical cord, and adipose tissues), and confirmed their strong anti-inflammatory, anti-fibrotic, and angiogenesis-remodeling capacities [22,26].

In our previous study, we demonstrated that the exposure of eMSC to an inflammatory environment upregulates their immunomodulatory transcriptional and inflammatory-/angiogenesis-related secretory profiles [12]. Consequently, MSC-derived exosomal cargos (EXOs) are a promising alternative to cellular therapeutics. In this study, EXOs isolated from Crude and CD146^+^ eMSC populations were characterized and compared for their miRNA therapeutic signatures. Furthermore, we assessed the capacity of eMSCs to affect macrophage and peripheral blood mononuclear cell functionality under inflammatory conditions. These types of observations could provide a rationale for further testing eMSC EXOs as a viable therapeutic modality to manufacture cell-free products for acute inflammation resolution, as well as in chronic conditions such as osteoarthritis and diabetes, where inflammation is increasingly recognized as an important component of disease progression.

## 2. Materials and Methods

### 2.1. Isolation, Culture, and Expansion of eMSC

Human endometrial tissue (*n* = 6) was collected according to our previous study [12] after participants provided written informed consent to the CryoVida stem cell bank (Guadalajara, Mexico). After eMSC expansion passage 0 (P0), cells were shipped to the University of Miami (Miami, FL, USA), then isolated, cultured, and expanded with complete Dulbecco’s Modified Eagle’s Medium plus 10% fetal bovine serum medium at 37 °C and 5% (*v*/*v*) CO_2_. All eMSCs were cultured by seeding 0.25 × 10^6^ cells/175 cm^2^ flask until 80% confluency until P3, and then detached with TrypLE Select Enzyme 1X (Gibco, Thermo Fisher Scientific, Waltham, MA, USA). We then assessed cell viability with 0.4% (*w*/*v*) Trypan Blue (Invitrogen, Thermo Fisher Scientific, Waltham, MA, USA). All tests were performed in accordance with the relevant guidelines and regulations following “not as human research” approval (based on the nature of the samples as discarded tissue).

### 2.2. Immunophenotype of eMSC

Flow cytometric analysis was performed on P3 eMSC (*n* = 3). Then, 2.0 × 10^5^ cells were labeled with antibodies specific for SUSD2 (BioLegend, San Diego, CA, USA) and CD146 (Miltenyi Biotec, Auburn, CA, USA), in addition to the corresponding isotype controls. All cells were stained with eFluor 780 fixable viability dye (Invitrogen). The fluorescent signal was acquired using a CytoFLEX S (20,000 events) and analyzed with Kaluza analysis software (Beckman Coulter, Miami, FL, USA).

### 2.3. CD146^+^ eMSC Selection

The eMSC original population (designated as the Crude group) was sorted based on CD146 expression to yield the CD146^+^ subpopulation. Briefly, Crude eMSCs were re-suspended in staining buffer containing PBS with 0.5% bovine serum albumin (BSA) and 2 mM EDTA, and then incubated with biotinylated anti-human CD146 (Miltenyi Biotech, Inc., Auburn, CA) at 4 °C for 20 min. The Invitrogen™ CELLection Dynabeads™ Biotin Binder Kit (Thermo Fisher Scientific) was used according to manufacturer’s instructions for magnetic selection.

### 2.4. Clonogenic Assay of Crude and CD146^+^ eMSCs

Crude and CD146^+^ eMSCs at passage three (P3) (*n* = 3) were seeded in 100-mm culture plates in triplicate, at a density of 10^3^ cells per plate in complete medium. Colony-forming unit fibroblasts (CFU-Fs) were manually enumerated on day 15 after cytochemical staining with 0.01% Crystal Violet (Sigma, Billerica, MA, USA).

### 2.5. Cell Proliferation Assay of Crude and CD146^+^ eMSCs

Crude and CD146^+^ eMSCs at passage three (P3) (*n* = 3) were seeded in a 24-well plate at a concentration of 1 × 10^3^ cells/well in triplicate, in complete medium. Growth curves were generated as percent confluency achieved on day 10 of culture, from bright-field images obtained using an IncuCyte live-cell analysis system with IncuCyte ZOOM software (Essen Bioscience, Ann Arbor, MI, USA).

### 2.6. Isolation and Validation of Crude and CD146^+^ eMSC EXOs

Crude and CD146^+^ eMSCs at passage three (P3) were seeded in complete medium until 70% confluency. Briefly, non-adherent cells were removed by Dulbecco’s phosphate buffered saline (DPBS; Sigma Aldrich, St. Louis, MO, USA). After gentle rinsing, an exosome-depleted medium was added to each group for 48 hrs. Conditioned media from each group cultured in an exosome-depleted medium were collected and centrifuged at 2000× *g* for 10 min to remove debris, then at 10,000× *g* for 30 min, and finally ultracentrifuged at 120,000× *g* for 16 h.

Pre-enriched exosome populations were incubated with the Dynabeads^®^-based Exosome-Human CD63 Isolation/Detection Reagent (Invitrogen) and purified according to the manufacturer’s instructions for magnetic selection. CD9 (Invitrogen) expression was used to validate exosome presence in CD63^+^-gated particles by flow cytometry. The specific fluorescent labeling of 20,000 events was analyzed on a CytoFLEX S with Kaluza analysis software (Beckman Coulter).

Nanoparticle tracking analysis (NTA) (NanoSight NS300, Malvern, Westborough, MA, USA) from Crude and CD146^+^ eMSC EXO was performed for quantity and size determination. All samples were diluted 1:50 in PBS. The following settings were set according to the manufacturer’s software: detection threshold 5; room temperature; number of frames 30; and measurement time 30 s. The size distribution and particle concentration each represent the mean of five individual measurements.

The functional assessment of Crude and CD146^+^ eMSC EXOs was performed by macrophage polarization and immunopotency assays. For EXO tracking of eMSCs, exosomes were stained with a PKH26 red fluorescent membrane staining kit (Fluorescent Cell Linker Kits, Sigma) according to the manufacturer’s instructions, and then co-cultured with target cells in functional assessments.

### 2.7. Quantitative Real-Time PCR (qPCR) of Crude and CD146^+^ eMSCs

A pre-designed 90-gene Taqman-based mesenchymal stem cell qPCR array (Stem Cell Technologies, Vancouver, Canada) was performed (*n* = 2) using 1000 ng cDNA per eMSC (Crude and CD146^+^) sample, and processed using a StepOne Real-time thermocycler (Applied Biosystems, Thermo Fisher Scientific, Waltham, MA, USA). Data analysis was performed using Stem Cell Technologies’ qPCR online analysis tools (Stem Cell Technologies). Sample and control Ct values were expressed as 2^−ΔΔ^Ct (with 38 cycles as the cut-off point). Expression levels were represented in bar plots ranked by transcript expression levels on a log-transformed scale of the sample cohort compared to the control. Bar plots were color-coded by the functional class of genes (namely Stemness, MSC, MSC-related/Angiogenic, Chondrogenic/Osteogenic, Chondrogenic, Osteogenic, and Adipogenic). A *t*-test (unpaired, two-tailed test with equal variance) was used in all statistical analysis, and *p*-values were corrected for multiple comparisons by the Benjamini–Hochberg procedure.

### 2.8. miRNA Profile of Crude and CD146^+^ eMSC EXOs

A total Exosome RNA and Protein Isolation Kit (Thermo Fisher Scientific) was used to extract miRNA from Crude and CD146^+^ eMSC EXOs, according to the manufacturer’s instructions. Total exosome miRNA (1 μg) was used for first-strand cDNA synthesis with the All-in-One miRNA First-Strand cDNA Synthesis Kit (GeneCopoeia, Rockville, MD, USA).

Pre-designed qPCR arrays covering 166 miRNA primers related to human MSC exosomes (GeneCopoeia) were performed using 1000 ng cDNA per Crude and CD146^+^ eMSC EXOs (*n* = 2), and then processed using a StepOne Real-time thermocycler (Applied Biosystems, LLC). The analysis was performed using GeneCopoeia’s online Analysis System (http://www.genecopoeia.com/product/qpcr/analyse/ (accessed on 20 May 2022)). Mean values were normalized with SNORD47 (a small nucleolar RNA) and expression levels were calculated using the 2^−ΔCt^ method.

A miRNet centric network visual analytics platform (https://www.mirnet.ca/ (accessed on 10 June 2022)) was used to created miRNA interactomes. The miRNA target gene data were collected from the well-annotated database miRTarBase v8.0. miRNA–gene interactome network refining was performed with 2.0 betweenness cut-off. Values (with a 34-cycle cut-off point) were represented in a topology miRNA–gene interactome network using a force atlas layout and hypergeometric test algorithm.

### 2.9. Macrophage Polarization Assay

Human monocytes (THP-1, ATCC) were differentiated into macrophages using PMA/IO (Phorbol 12-myristate 13-acetate/Ionomycin) and polarized to M1 macrophages by a M1-macrophage generation medium (PromoCell, Heidelberg, Germany). Then, 5.0 × 10^4^ PMA/IO-stimulated THP-1 (macrophages) were mixed with Crude or CD146^+^ eMSC EXOs (*n* = 3 for each) per well of a 24-well plate and cultured in M1-macrophage generation medium for 2 days. Macrophage polarization status was assessed using a polarization qPCR array (ScienCell, Carlsbad, CA, USA).

RNA extraction from THP-1 cultures was performed using the RNeasy Mini Kit (Qiagen, Frederick, MD, USA) according to the manufacturer’s instructions. Total RNA (1 μg) was used for reverse transcription with a SuperScript™ VILO™ cDNA synthesis kit (Invitrogen). A pre-designed 40-gene Human Macrophage Polarization array (GeneQuery™ Human Macrophage Polarization Marker qPCR Array Kit, ScienCell) was performed using 1000 ng cDNA per culture and processed using a StepOne Real-time thermocycler (Applied Biosystems). Mean values were normalized to GAPDH, and expression levels were calculated using the 2^−ΔCt^ method. Values were represented in a stacked bar plot for M0, M1, and M2 polarization as the relative fold change of the PMA/IO + THP-1/eMSC EXOs to PMA/IO + THP-1 (reference sample, 2^−ΔCt^ = X sample/X reference sample).

The miRDB online database (http://mirdb.org (accessed on 20 June 2022)) for prediction of functional microRNA targets has been used to correlate highly expressed target genes in macrophages with specific miRNAs identified by eMSC EXO miRNA profiling. MirTarget prediction scores are in the range of 0–100% probability, and candidate transcripts with scores ≥ 50% are presented as predicted miRNA targets in miRDB [27].

### 2.10. Phagocytosis Assay

For this step, 9.0 × 10^3^ PMA/IO-stimulated THP-1 (macrophages) were mixed with Crude or CD146^+^ eMSC EXOs (*n* = 2 for each) per well of a 96-well plate, and cultured in M1-macrophage generation medium for 2 days. Parallel wells were designed as negative (M1 medium only) and positive (PMA/IO-stimulated THP-1 only) controls. On day 2, all cultures were incubated with 1.0 mg/mL fluorescein-labeled Escherichia coli K-12 bioparticles at 37 °C and 5% CO_2_ for 2 h, then cytochemical stained with 1.25 mg/mL Trypan Blue according to the manufacturer’s instructions (Thermo Fisher Scientific). Levels of phagocytosed fluorescent bioparticles were determined at 480 nm and 520 nm (SpectraMax M5 spectrophotometer, Molecular Devices, San Jose, CA, USA) and quantified using the following formula: %fluorescence = (net experimental reading/net positive control reading) ∗ 100.

### 2.11. Immunopotency Assay (IPA)

An immunopotency assay was performed similarly to that in [17]. Briefly, 3.0 × 10^5^ CFSE-labeled human peripheral blood mononuclear cells (PBMC, two male and two female donors, Stem Cell Technologies) were mixed with Crude or CD146+ eMSC EXOs (*n* = 3 for each) per well of a 24-well plate in MLR culture medium. The MLR medium contained RPMI 1640 + GlutaMAX (Invitrogen) supplemented with 15% heat inactivated normal human AB serum (Corning, Corning, NY, USA), 1% penicillin-streptomycin, 1% non-essential amino acids, 1% sodium pyruvate, 1% vitamins (Invitrogen), and 2 mM HEPES (Corning Mediatech, NY). Phorbol myristate acetate and ionomycin (PMA/IO 500X, ThermoFisher Scientific) were added to stimulate PBMC proliferation, with unstimulated cells used as controls. Cultures were incubated at 37 °C and 5% CO_2_ for 4 days, harvested, and stained with viability dye (Live/Dead) (ThermoFisher Scientific). A CFSE^+^ proliferation signal was acquired using a CytoFLEX S (20,000 events) and analyzed with Kaluza analysis software (Beckman Coulter).

### 2.12. Protein Profile of Cells/eMSC EXOs Co-Cultures Secretome

Multiplex protein arrays of 60 cytokines (RayBio^®^ C-Series, RayBiotech, Peachtree Corners, GA, USA) were used to determine THP-1/eMSC EXO and PBMC/eMSC EXO co-culture secretomes (*n* = 2). For each assay, 1 mL of co-culture supernatant was used, following the manufacturer’s instructions. The data shown represent 40 s of exposure in the FluorChem E chemiluminescence imaging system (ProteinSimple, San Jose, CA, USA). Results were generated by quantifying the mean spot pixel density of each array using a protein array analyzer plugin coupled with ImageJ software (Fiji/ImageJ, NIH website). Signal intensities were normalized with the background, and separate signal intensity results represent the average pixel density of two spots per protein. The signal intensity for each protein spot is proportional to the relative concentration of the antigen in the sample.

### 2.13. Statistical Analysis

A normal distribution of values was assessed by the Kolmogorov–Smirnov normality test. Statistical analysis was performed using paired and unpaired Student’s *t*-test for normally distributed data with Wilcoxon (for paired data) or Mann–Whitney (for unpaired data) tests. One-way ANOVA was used for multiple comparisons. All tests were performed with GraphPad Prism v7.03 (GraphPad Software, San Diego, CA, USA). The level of significance was set at *p* < 0.05.

## 3. Results

### 3.1. Crude and CD146^+^ eMSC Characterization

Our preceding study confirmed that eMSCs share phenotypic surface markers with pericytes (CD146, CD140b and NG2) and the expression of SUSD2 [12], which is consistent with others [3]. On this basis, eMSCs were expanded until P3 and the co-expression of CD146^+^ and SUSD2 was evaluated, demonstrating positive expression levels of 52.97 ± 17.85% between markers. Additionally, eMSCs showed expression levels of 21.74 ± 17.56% for CD146^+^/SUSD2^-^ and 16.87 ± 6.09% for SUSD2^+^/CD146^-^ (Figure 1A). Following positive selection, CD146^+^ and Crude eMSC were characterized by standard assays to evaluate the quality of both groups, showing similar morphology, growth kinetics (Figure 1C), and clonogenic capacity (Figure 1B), although CD146^+^ eMSCs displayed slightly higher colony sizes (Figure 1B).

Molecular profiling of CD146^+^ eMSCs versus Crude eMSCs revealed that 55 out of 90 genes tested were more highly expressed in CD146^+^ eMSCs, with 21 genes being more than two-fold higher (*VCAM1, PPARG, CSPG4, IL6, VEGFA, MCAM, BDNF, LIF, CD200, DLX2, ACAN, HIC1, FGF2, PDGFRB, COL1A1, TWIST2, NOTCH1, EGF, KITLG, GDF15, FUT4*) (Figure 1D). Interestingly, the tested genes were grouped into phenotype/function-related cohorts, with the MSC-associated, stemness, and MSC cohorts showing the most prominent fold expression change overall between CD146^+^ eMSC and Crude eMSC cultures (Figure 1D). The *VCAM-1* gene, whose expression was 16-fold higher in CD146^+^ eMSCs, is directly related to robust pro-angiogenic and immunosuppressive MSC actions. From the highly expressed genes (more than three-fold), *PPARG*, *IL6,* and *BDNF* have important roles in immunoregulation and cell apoptosis. *PPARG* regulates the expression of genes involved in the DNA damage response of inflamed endometrium [28]. In addition, *IL6* has an intracellular role in MSC immunosuppression and proliferation, as its high expression is related to the increased capacity of MSCs to suppress activated T-cell proliferation [29]. In the same context, BDNF gene expression has pro-survival effects, and regulates intracellular signaling molecules in order to inhibit inflammatory cytokine expression in MSC [30].

In contrast, Crude eMSC molecular profiles showed higher expression of genes involved in MSC differentiation programs towards adipogenic, chondrogenic, and osteogenic lineages, with nine genes (*FGF10, CEBPA, TERT, FGF18, SP7, ALPL, KDR, BMP6, VWF*) being more highly expressed by over two-fold compared to CD146^+^ eMSCs. Specifically, the *FGF10, CEBPA,* and *TERT* genes seem to be a characteristic molecular signature for Crude eMSCs, as they are expressed 225-, 112-, and 95-fold more highly, respectively. These genes are involved in the proliferation and differentiation of MSC signaling. Specifically, studies showed that *FGF10* expression and protein paracrine secretion control epithelial proliferation and ligand–receptor signaling in the endometrium [31]. Additionally, *CEBPA* and *TERT* gene expressions increase MSCs’ stem-like properties and proliferation potential [32,33].

### 3.2. Crude and CD146^+^ eMSC EXOs Characterization

Upon ultracentrifugation and CD63^+^ immunoselection, CD9, a typical surface marker used to validate exosome presence by flow cytometry, was analyzed, and demonstrated high purity (>90%) in Crude (Figure 2A) and CD146^+^ eMSC EXOs (Figure 2B). Additionally, EXOs were characterized by size using a nanoparticle tracking analysis (NTA). Both crude (Figure 2A) and CD146^+^ (Figure 2B) eMSC EXOs showed sizes < 200 nm.

### 3.3. miRNA Profile of Crude and CD146^+^ eMSC EXOs

From the 166 MSC-related miRNAs analyzed, 121 and 88 miRNA cargos were present in Crude (Figure 2A) and CD146^+^ eMSC EXOs (Figure 2B), respectively, with 82 MSC-related miRNAs commonly present in both populations. Interestingly, three miRNAs were highly present (>10^0^ relative expression to *SNORD47*) in both Crude and CD146^+^ eMSC EXOs, namely hsa-miR-107, hsa-miR-125a, and hsa-miR-301a-3p. All three miRNAs showed strong involvement in immune cell regulation, indicating the strong immunomodulatory actions of eMSC EXOs.

In general, from the commonly present miRNAs in both populations, four miRNAs were more enriched in CD146^+^ eMSC EXOs, including hsa-miR-320e, hsa-miR-182-3p, hsa-miR-378g, and hsa-let-7e-5p. Among these molecules, hsa-miR-182-3p has been associated with reduction in cell apoptosis, reduction in pro-inflammatory cytokine expression, and attenuation of the inflammatory response via IKKβ/NF-κB modulation [34].

Reactome and KEGG analysis of miRNAs highly present in Crude and CD146^+^ eMSC EXOs indicated their involvement in the regulation of gene expression, immune system, cell cycle, cellular responses to stress, cytokine signaling, and MAPK signaling pathways (Figure 3 and Figure 4). Furthermore, miRNA–gene interactome network analysis revealed four miRNAs (hsa-mir-21-5p, hsa-mir-32-5p, hsa-mir-98-5p, and hsa-let-7e-5p) for Crude eMSC EXOs and three miRNAs (hsa-mir-27b-3p, hsa-mir-98-5p, and hsa-let-7e-5p) for CD146^+^ eMSC EXOs with higher node degrees that act as hubs in the gene network. Even though the levels of these miRNAs as cargo within the eMSC EXOs are variable, they regulate multiple genes related to important signaling pathways.

### 3.4. Crude and CD146^+^ eMSC EXOs Effects on Macrophages

In co-cultures, PMA/IO-stimulated THP-1 internalized eMSC EXOs (Figure 5A). At the molecular level, PMA/IO-stimulated THP-1 molecular profiling indicated a strong gene expression shift towards an M2 macrophage polarization by both Crude and CD146^+^ eMSC EXOs (Figure 5B). Most importantly, the expression levels of *MRC1* (*CD206*), a characteristic M2-polarization macrophage marker [35], was strongly induced when macrophages were exposed to CD146^+^ eMSC EXOs. *TGFB1* and *CCL2* expression levels were also significantly induced by CD146^+^ eMSC EXOs. These three highly expressed genes in M2-polarized macrophages upon exposure to eMSC EXOs were used for in silico correlation with identified specific miRNA cargo during eMSC EXOs miRNA profiling. Interestingly, a MirTarget prediction scoring system revealed *MRC1* as a target for hsa-miR-1255a and hsa-miR-3065-5p (60% and 71% probability, respectively), *TGFB1* as a target for hsa-let-7e-5p (59%), and *CCL2* as a target for hsa-miR-125a (51%) (Figure 5C). Notably, all four miRNAs were highly present in both Crude and CD146^+^ eMSC EXOs. eMSC EXOs significantly increased secretion of M2-polarization-related molecules (CCL2, GRO, and TIMP-2) in PMA/IO-stimulated THP-1 compared to stimulated THP-1 alone (Figure 5D).

Functionally, PMA/IO-stimulated THP-1 exposed to Crude or CD146^+^ eMSC EXOs for 2 days showed reduced capacity to phagocytize fluorescent bioparticles (Figure 5E). In contrast, PMA/IO-stimulated THP-1 monocultures showed increased phagocytosis capacity in vitro. According to previous studies, high levels of phagocytic activity are directly related to strong polarization of macrophages towards the M1 pro-inflammatory phenotype, especially during acute inflammation [36]. In contrast, M2-like macrophage polarization resulted in their reduced phagocytosis capacity. In pathogen-free inflammation, this phenomenon can be related to removal of the remaining apoptotic cells at the final stages of inflammation, when macrophages have already polarized towards the M2 phenotype [37].

### 3.5. Crude and CD146^+^ eMSC EXOs Effects on Peripheral Blood Mononuclear Cells

CFSE-labeled human naïve PBMC showed an average proliferation of ~25.4% for male and female PBMC donors. Exposure of naïve PBMC in eMSC EXOs showed no effect on their proliferation capacity (Figure 6A, upper panel, and Figure 6B). Upon PMA/IO stimulation, PBMC showed an average proliferation of ~74.2% for male and female PBMC donors. Exposure of PMA/IO-stimulated PBMC to two different eMSC EXOs concentrations (EXOs and 2x EXOs) resulted in minimal suppression of PBMC proliferation. Interestingly, both Crude and CD146^+^ eMSC EXOs showed similar suppression capacity of the stimulated PBMC proliferation: on average, 1.4% suppression for both male and female PBMC donors (Figure 6A, bottom panel, and Figure 6B).

As eMSC EXOs showed no significant effect on stimulated PBMC proliferation, we further investigated their inflammation-related cytokine profiles upon eMSC EXOs exposure. Overall, both Crude and CD146^+^ eMSC EXOs exposure resulted in reduced secretion of pro-inflammatory cytokines compared to stimulated PBMC alone. Specifically, 13 (GM-CSF, ICAM-1, IFN-γ, IL-2, IL-6 IL-6 sR, IP-10, MCP-1, MIP-1α, MIP-1β, RANTES, sTNF RII, and TIMP-2) out of 40 molecules tested had significantly (*p <* 0.05) lower secretion in eMSC EXOs + stimulated PBMC cultures. To assess the relationship between the proteins identified, a protein association network analysis was performed using STRING 11.0 software. According to a K-means clustering algorithm, all 13 proteins showed high protein–protein interaction (PPI) enrichment (*p <* 1.0 × 10^−16^) and an average local clustering coefficient >0.9, indicating that they were strongly biologically connected. Importantly, five biological processes were significantly enriched in the protein network, namely the cytokine-mediated signaling pathway (GO:0019221), positive regulation of the immune system process (GO:0002684), leukocyte activation (GO:0045321), positive regulation of leukocyte migration (GO:0002687), and monocyte chemotaxis (GO:0002548) (Appendix A). Additionally, five KEGG and reactome pathways were significantly enriched in the protein network, namely the TNF signaling (hsa:04668), Toll-like receptor signaling pathway (has:04620), JAK-STAT signaling pathway (has:04630), Chemokine signaling pathway (has:04062), and Signaling by Interleukins (HAS:449147) (Appendix A). Overall, lower secretion of these 13 pro-inflammatory proteins upon exposure of PBMC to eMSC EXOs indicated the strong immunomodulatory effect of eMSC EXOs on PBMC activation and inflammatory signaling in vitro.

In parallel, two anti-inflammatory proteins, IL-10 and IL-13, showed higher secretion, especially in CD146^+^ eMSC EXOs + stimulated PBMC compared to stimulated PBMC alone cultures. Importantly, both IL-10 and IL-13 play crucial roles in macrophage polarization towards the M2 phenotype by upregulating the expression of arginase 1 (Arg1) and CD206 M2-polarization macrophage markers [38,39]. Collectively, from these data as well as those from the macrophage polarization assay, we can conclude that both Crude and CD146^+^ eMSC EXOs show strong M2 macrophage polarization effects in vitro.

## 4. Discussion

The human endometrium has emerged as an attractive source of mesenchymal stem/stromal cells (eMSC) that are easily isolated by non-invasive procedures, and show increased immunomodulatory and pro-angiogenic properties [12]. Furthermore, we have demonstrated that the CD146 signature is correlated with innately higher MSC immunomodulatory and secretory capacity, and, thus, better therapeutic potency in vivo [9]. Interestingly, at the extracellular vesicle level, our previous studies clearly demonstrated that infrapatellar fat pad-derived (IFP) MSCs show a potent miRNA immunomodulatory exosomal (EXOs) profile. Functionally, IFP-MSC EXOs can significantly affect macrophage polarization under inflammatory conditions both in vitro and in vivo by inducing macrophages towards an anti-inflammatory therapeutic M2 phenotype [22]. Herein, for the first time, we elucidated the miRNA exosomal profile of Crude and CD146^+^ eMSCs. Our findings provide critical information on the immunomodulatory effects of eMSC EXOs on macrophages’ and peripheral blood mononuclear cells’ functionality. These types of investigations could provide a rationale for further testing of eMSC EXOs as a viable therapeutic modality to manufacture cell-free products for inflammatory conditions, including osteoarthritis and diabetes. Specifically, macrophage infiltration and pro-inflammatory activation has been associated with synovitis/fat pad fibrosis severity in the knee joints of osteoarthritis patients [40,41], as well as pancreatic islet viability/insulin secretion capacity in diabetes patients [42,43].

SUSD2 was identified as a single marker capable of purifying eMSCs possessing MSC properties, and confirmed that these cells reside in a perivascular niche [3]. Consistent with previous studies [3,10], herein, we showed that SUSD2^+^ eMSC partially co-express the CD146 pericytic marker. According to our findings, the CD146^+^ eMSC subpopulation shows a distinct molecular profile that is directly related to immunosuppressive, pro-angiogenic, and anti-apoptotic MSC actions. Specifically, increased *VCAM-1* expression in CD146^+^ eMSC is a potent mechanism by which MSCs exert their immunosuppressive effects via increased cell–cell adhesion with T cells [44]. Studies have shown that VCAM-1^+^ MSCs possess a favorable angiogenic paracrine activity, and display therapeutic potential in vascular ischemia animal models [45]. In contrast, the Crude eMSC molecular profile is mainly characterized by high expression of genes involved in MSC differentiation programs. Together, our data indicate that CD146^+^ eMSCs possess a superior immunomodulatory, pro-angiogenic, pro-survival molecular profile compared to Crude eMSCs.

Upon eMSC EXOs purification from Crude and CD146^+^ eMSC populations, hsa-miR-107, hsa-miR-125a, and hsa-miR-301a-3p miRNAs involved in immune system regulation were highly present in their exosomes. Specifically, the expression levels of hsa-miR-107 have been demonstrated to be related to TLR4 activation, whereas decreased expression of hsa-miR-107 may be a regulative feedback effect to limit insulin resistance in inflammation [46]. In addition, hsa-miR-125a stabilizes Treg-mediated immune homeostasis [47], whereas hsa-miR-301a-3p induces the M2 polarization of macrophages via activation of the PTEN/PI3Kγ signaling pathway [48]. Furthermore, we demonstrated that these immunomodulatory exosomal signatures can effectively induce PMA/IO-stimulated macrophages to polarize towards an anti-inflammatory therapeutic M2 phenotype. Notably, we observed that exposure of stimulated macrophages to CD146^+^ eMSC EXOs resulted in a more robust polarization towards M2-like macrophages by upregulation of the *MRC1, TGFB1,* and *CCL2* genes. Studies have shown that TGF-β induces M2-like macrophage polarization via SNAIL-mediated suppression of a pro-inflammatory phenotype [49]. CCL2 is associated with monocyte recruitment in inflamed tissues via the CCR2 chemokine receptor after pro-inflammatory cytokine activation. Importantly, CCL2 and CCR2 determine the extent of M2 macrophage polarization by enhancing the production of the anti-inflammatory IL-10 cytokine [50].

In silico prediction analysis for *MRC1, TGFB1,* and *CCL2* gene interaction with identified miRNA cargos from miRNA EXOs profiling revealed four miRNAs (hsa-let-7e-5p, hsa-miR-125a, hsa-miR-1255a, and hsa-miR-3065-5p) that strongly modulate their expression. Previous studies have demonstrated that the let family miRNAs may regulate M2 polarization through the SOCS1/STAT pathway [51]. Additionally, hsa-miR-125a affects monocyte adhesion and chemotaxis by direct targeting of the chemotaxis-mediating chemokine receptor CCR2 [52], whereas hsa-miR-1255a can regulate SMAD4 to participate in the TGF-β signaling pathway [53]. However, little is known about the biological function of hsa-miR-3065-5p on macrophages, and, therefore, it requires further investigation. Functionally, the pronounced M2-like macrophage polarization action by eMSC EXOs has been clearly demonstrated by the reduced phagocytic activity of macrophages. This may suggest that macrophage exposure to eMSC EXOs not only alters their molecular profile and phenotype, but may also significantly affect their functionality during inflammatory conditions in vivo.

At the cellular level, we have previously demonstrated that the immunomodulatory potential of eMSC is specifically related to their strong inhibitory effect on PBMC proliferation [12]. Along the same lines, Queckbörner et al. reported that eMSC co-culturing with PBMC can effectively suppress the proliferation and activation of CD4^+^ T cells [54]. In the present study, we investigated the effect of Crude and CD146^+^ eMSC EXOs on PBMC proliferation and pro-inflammatory secretory activity. Importantly, our data indicate that even though eMSC EXOs show a privileged immunomodulatory miRNA profile, they cannot significantly affect the proliferation of stimulated PBMC. To the best of our knowledge, this is the first time that such an effect has been reported. This finding may be attributed to the mechanisms used by MSC to suppress activation and proliferation of PBMC in vivo. In general, MSCs’ immunoregulatory function requires their preliminary activation by immune cells through local secretion and stimulation by pro-inflammatory molecules, such as IFNγ, TNFα, IL-1α, and IL-1β [55]. In turn, MSCs activate their immunosuppressive and anti-inflammatory responses, mediated by several soluble factors, including IDO, PGE2, transforming growth factor β (TGFβ), insulin-like growth factor (IGF), and interleukin 10 (IL-10) [56,57,58]. Therefore, an absence of these important immunomodulatory soluble factors from eMSC EXOs can justify their limited capacity to suppress PBMC proliferation. However, the strong immunomodulatory attributes of eMSC EXOs are evident by the acquisition of a reduced pro-inflammatory secretory profile by stimulated PBMC. Specifically, we confirmed that 13 (GM-CSF, ICAM-1, IFN-γ, IL-2, IL-6 IL-6 sR, IP-10, MCP-1, MIP-1α, MIP-1β, RANTES, sTNF RII, and TIMP-2) inflammation-related cytokines had significantly lower secretion in eMSC EXOs + stimulated PBMC cultures compared to stimulated PBMC alone. In parallel, PBMC exposure to eMSC EXOs induces their higher secretion of the major anti-inflammatory proteins IL-10 and IL-13. IL-10 is a potent anti-inflammatory cytokine that inhibits MHC class II and costimulatory molecule B7-1/B7-2 expression on monocytes and macrophages, and limits the production of pro-inflammatory cytokines (including IL-1α and β, IL-6 and TNF-α) and chemokines (IP-10, MCP-1, and RANTES) [59]. Similarly, IL-13 is a strong anti-inflammatory cytokine, produced by T helper 2 cells and naïve or memory CD4^+^/CD8^+^ T cells, that can inhibit the secretion of pro-inflammatory mediators, including nitric oxide (NO), IL-1β, IL-6, IL-12, and TNF-α [38]. Overall, both Crude and CD146^+^ eMSC EXOs induce a strong suppression of stimulated PBMC pro-inflammatory secretory activity.

## 5. Conclusions

In summary, eMSCs possess a potent miRNA immunomodulatory exosomal profile. Specifically, the CD146^+^ eMSC subpopulation demonstrates a significantly reinforced anti-inflammatory molecular profile compared to that of Crude eMSC, an effect that is reflected in their differential miRNA EXOs signatures. Functionally, eMSC EXOs, and foremost, CD146^+^ eMSC EXOs, significantly affect macrophage and peripheral blood mononuclear cell functionality under inflammatory conditions in vitro. On this basis, our results help to elucidate the various local therapeutic anti-inflammatory activities of eMSC EXOs.

## Figures and Tables

**Figure 1 cells-11-04002-f001:**
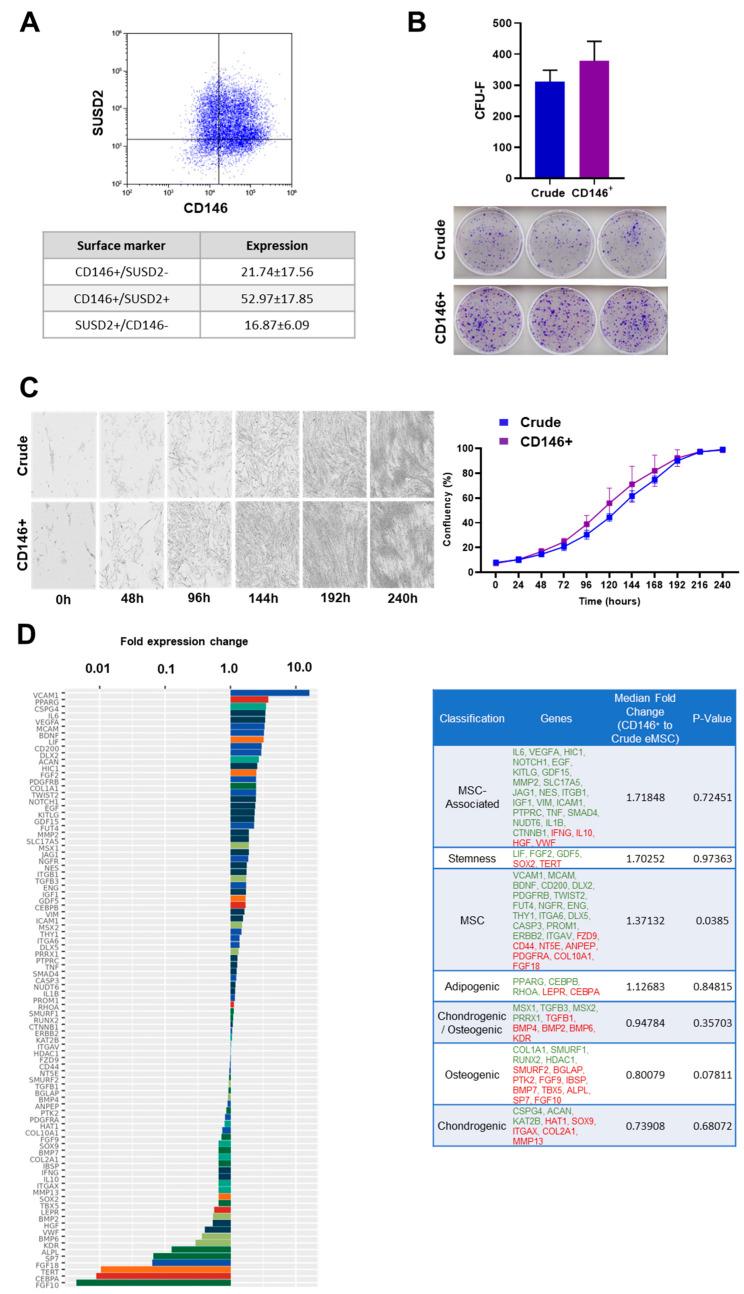
Crude and CD146^+^ eMSC immunophenotype, clonogenicity, growth kinetics, and molecular profiling. (**A**) eMSCs show high co-expression levels for CD146 and SUSD2 markers. (**B**,**C**) Crude and CD146^+^ eMSCs show similar morphology, clonogenic capacity, and growth kinetics. (**D**) Molecular profiling of CD146^+^ eMSC versus Crude eMSC cultures revealed that 55 out of 90 genes tested were expressed higher in CD146^+^ eMSCs, with 21 genes being more than two-fold higher. In the table, tested genes were grouped in phenotype/function-related cohorts. Green highlighted genes show higher expression, and red highlighted genes show lower expression.

**Figure 2 cells-11-04002-f002:**
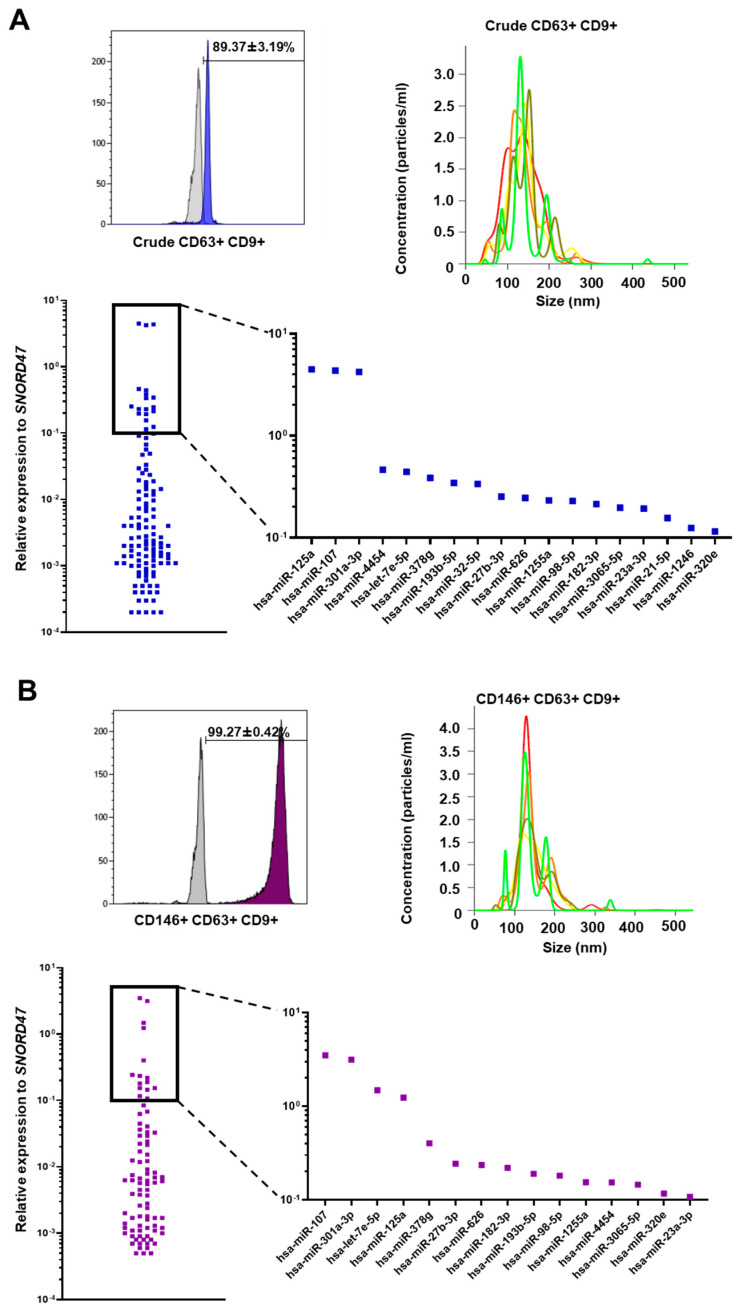
Crude and CD146^+^ eMSC EXOs characterization. (**A**,**B**) Nanoparticle tracking analysis showed that isolated vesicles from eMSC-conditioned media are <200 nm in diameter, consistent with the known size of exosomes. CD63^+^-selected exosomes showed purity by high positivity for exosome marker CD9. A total of 121 and 88 miRNA cargos were present in Crude and CD146^+^ eMSC EXOs, respectively.

**Figure 3 cells-11-04002-f003:**
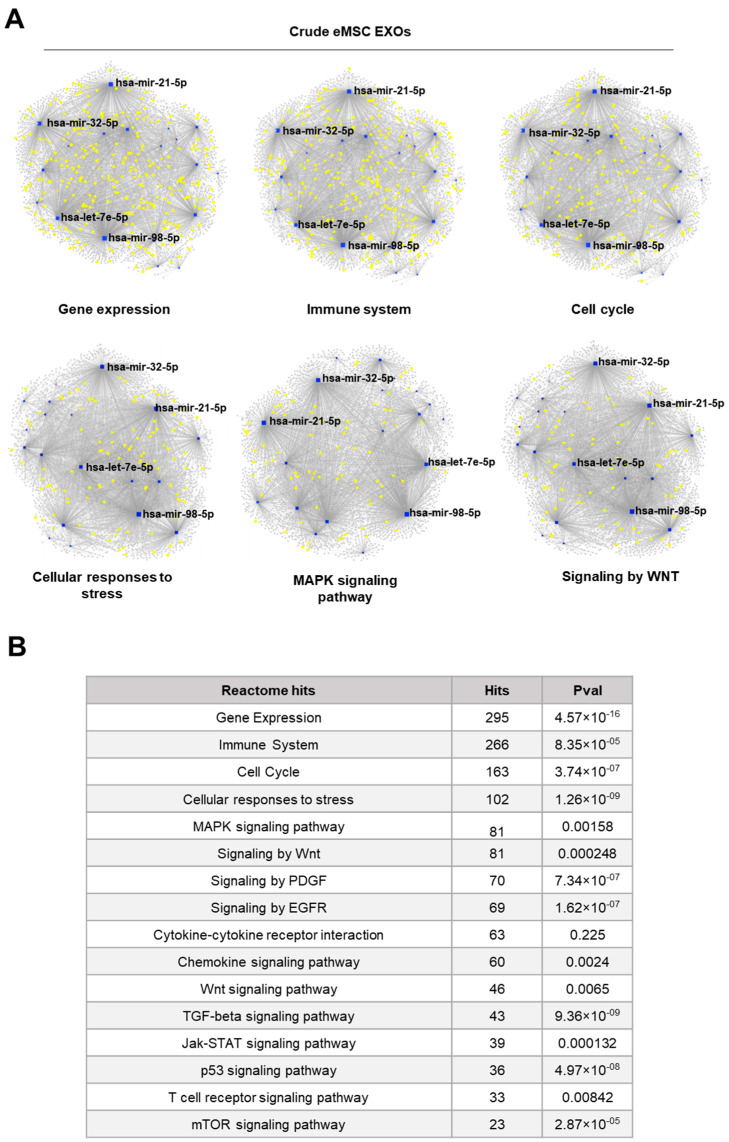
Reactome and KEGG analysis of miRNAs highly present in Crude eMSC EXOs. (**A**,**B**) Four miRNAs (hsa-mir-21-5p, hsa-mir-32-5p, hsa-mir-98-5p, and hsa-let-7e-5p) for Crude eMSC EXOs with higher node degrees act as hubs in the gene network. Crude eMSC EXOs showed their involvement in the regulation of six gene groups related to gene expression, immune system, cell cycle, cellular responses to stress, MAPK signaling, and WNT signaling pathways.

**Figure 4 cells-11-04002-f004:**
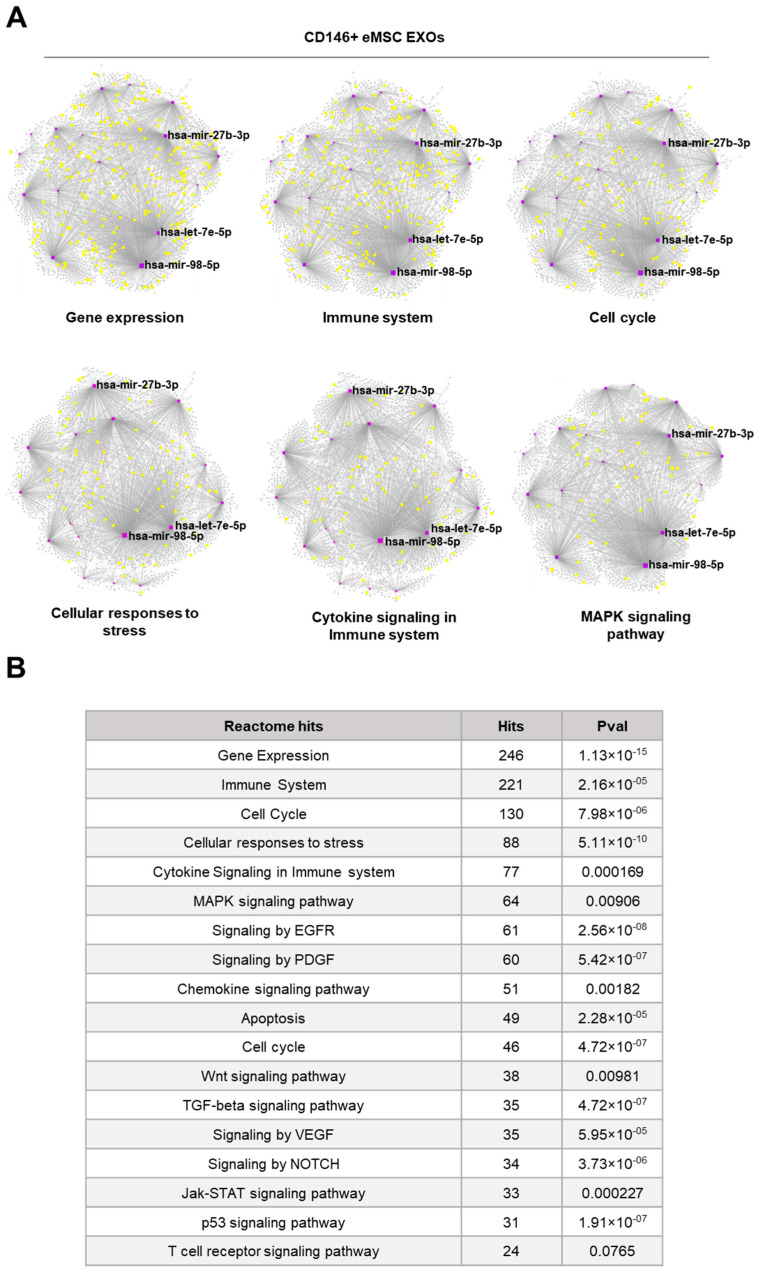
Reactome and KEGG analysis of miRNAs highly present in CD146^+^ eMSC EXOs. (**A**,**B**) Three miRNAs (hsa-mir-27b-3p, hsa-mir-98-5p, and hsa-let-7e-5p) for CD146^+^ eMSC EXOs with higher node degrees act as hubs in the gene network. CD146^+^ eMSC EXOs showed their involvement in the regulation of six gene groups related to gene expression, immune system, cell cycle, cellular responses to stress, cytokine signaling in immune system, and MAPK signaling pathways.

**Figure 5 cells-11-04002-f005:**
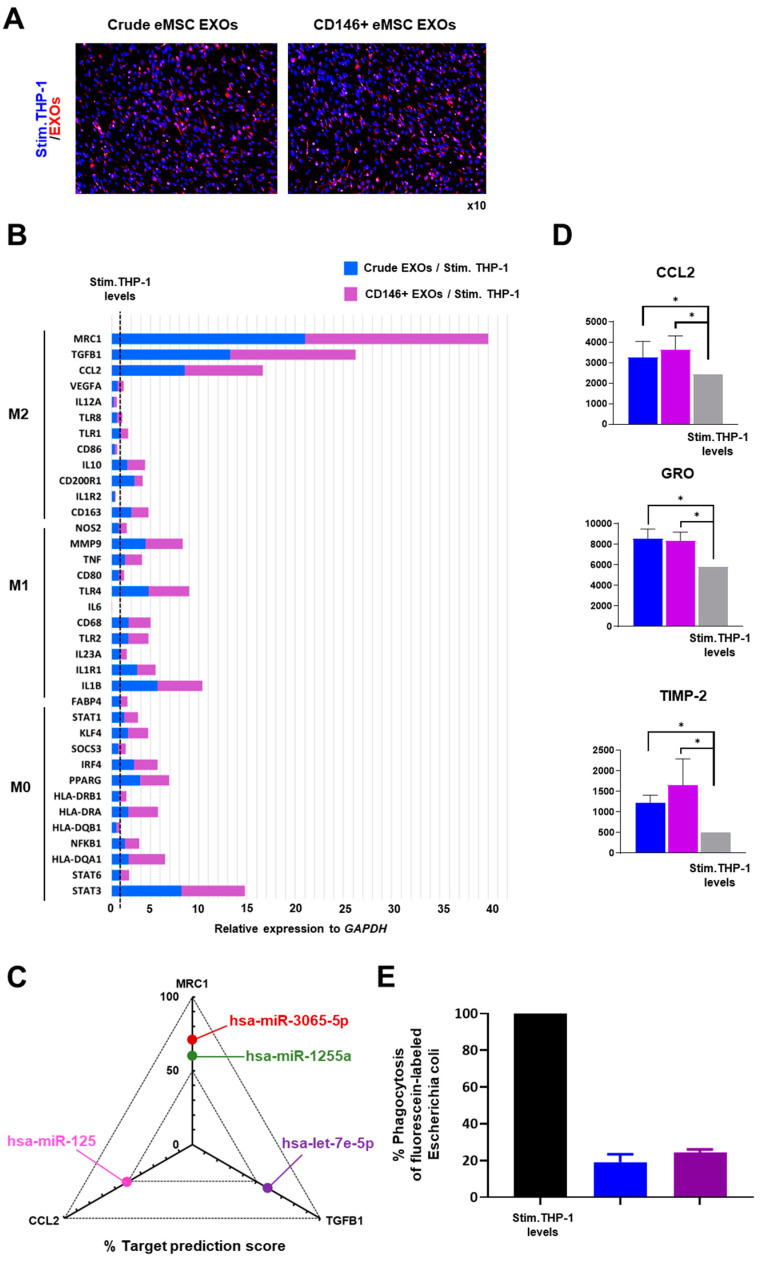
eMSC EXOs immunomodulatory effects on macrophages. (**A**) Both Crude and CD146^+^ eMSC EXOs were internalized in PMA/IO-stimulated THP-1 (blue, nucleus; red, EXOs). (**B**) PMA/IO-stimulated THP-1 molecular profiling indicated a strong shift in genes towards M2 macrophage polarization by both Crude and CD146^+^ eMSC EXOs. The dotted line represents the stimulated THP-1 gene levels without exposure to eMSC EXOs. (**C**) The *in silico* prediction scoring system revealed MRC1 as a target for hsa-miR-1255a and hsa-miR-3065-5p (60% and 71% probability, respectively), TGFB1 as a target for hsa-let-7e-5p (59%), and CCL2 as a target for hsa-miR-125a (51%). All four miRNAs were highly present in both Crude and CD146^+^ eMSC EXOs. (**D**) eMSC EXOs significantly increased secretion of M2-polarization-related molecules in PMA/IO-stimulated THP-1 compared to stimulated THP-1 alone (*, *p <* 0.05). (**E**) Crude and CD146+ eMSC EXOs exposure of PMA/IO-stimulated THP-1 resulted in reduced capacity to phagocytize fluorescent bioparticles (19 ± 4% and 24 ± 2%, respectively).

**Figure 6 cells-11-04002-f006:**
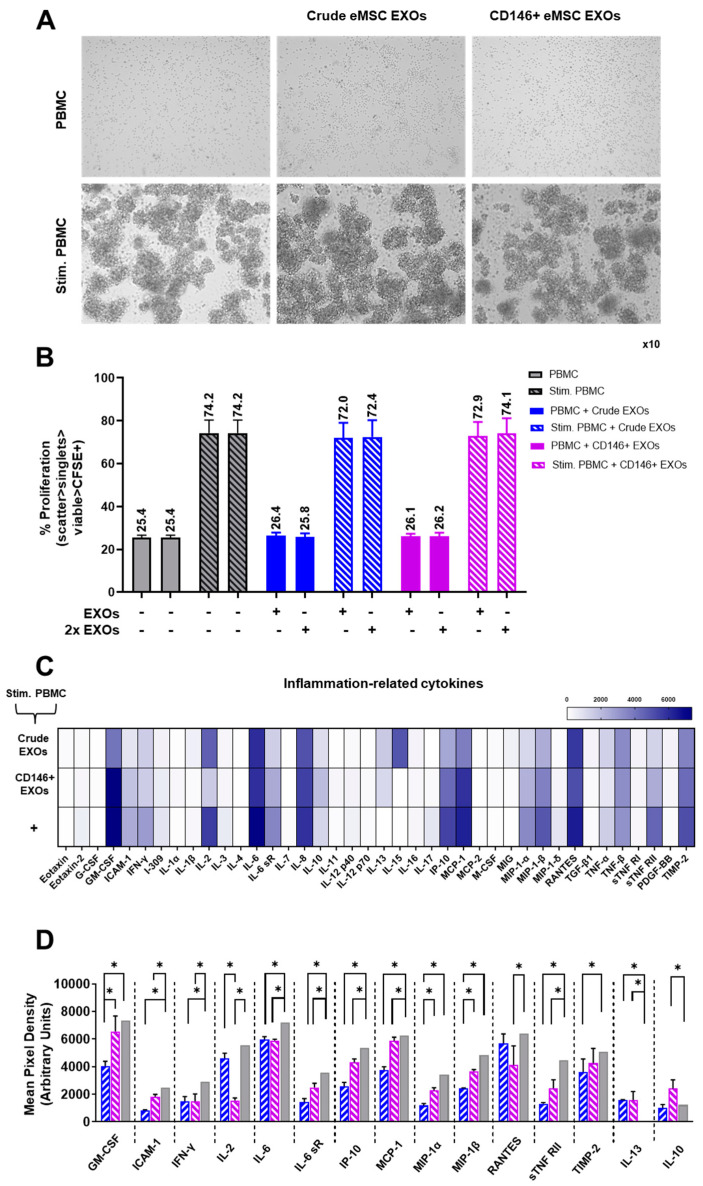
eMSC EXOs immunomodulatory effects on PBMC. (**A**) Unstimulated PBMC shows unaltered morphology with and without eMSC EXOs exposure (upper panel). Stimulation of PBMC with PMA/IO resulted in their characteristic aggregation in vitro, which was not affected by exposure to eMSC EXOs (bottom panel). (**B**) The percentage of CFSE-labeled PMA/IO-activated PBMC proliferation (relative to unstimulated PBMC) was measured in PBMC after exposure to Crude and CD146^+^ eMSC EXOs. Exposure of PMA/IO-stimulated PBMC to two different eMSC EXOs concentrations (EXO and 2x EXO) resulted in insignificant PBMC proliferation suppression. Every eMSC EXOs donor (Crude or CD146^+^, *n* = 3) was co-cultured in independent experiments with four different PBMC populations obtained from unrelated donors. (**C**) Both Crude and CD146^+^ eMSC EXOs exposure resulted in reduced secretion of pro-inflammatory cytokines compared to stimulated PBMC alone (marked as +). (**D**) A total of 13 out of 40 molecules tested had significantly (*p <* 0.05) lower secretion, and 2 out of 40 molecules tested had significantly (*, *p* < 0.05) higher secretion in eMSC EXOs + stimulated PBMC cultures compared to stimulated PBMC alone (blue bar: Stim. PBMC + Crude EXOs, pink bar: Stim. PBMC + CD146+ EXOs, grey bar: Stim. PBMC).

## Data Availability

Not applicable.

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
