# Peer review of "CD146+ Endometrial-Derived Mesenchymal Stem/Stromal Cell Subpopulation Possesses Exosomal Secretomes with Strong Immunomodulatory miRNA Attributes"

_cells, 2022, doi:10.3390/cells11244002_

Round 1

Reviewer 1 Report

Dear Authors,

Please find my comments in the attached PDF file.

General considerations:

The paper is quite hard to follow. The intro switches between MSC and eMSC and it's quite confusing. Figures are not cited in the text so I had to go around and search to see where the info is presented. More attention needs to be spent for these details as this sends a message of carelessness which makes one think whether the same attention was given to the experiments in general.

Thp1 cell line under PMA differentiates in macrophages. Please make sure in your results that this aspect is clear. It's not easy to understand when your experiments were with Thp1 or with Thp1/Mfs.

The whole article makes comparisons between mixed (crude) population of eMSCs and CD146+ eMSC. From the graphical abstract one gets the idea that you sort the population into CD146+ and negative cells. But apparently this is not the case... Please clarify this issue. If the crude population is indeed the original one that included also CD146+ then it's indeed obvious that the crude population has similar results as the CD146+ one. Moreover, based on the result from figure 1A, it looks like most of the population is CD146+. Please comment on this. Also the graphical abstract should be changed so to show that the crude pupulation included the CD146+ one.

I believe the graphical abstract should include the most important findings of the paper. If it gets too busy with too many elements you can separate the info and make the graphical abstract with your results (for example the M2 polarization is an important finding, the deregulated pathways, the top miRs etc) and make something similar to the original graphical abstract as a study "workflow".

Author Response

Reviewer 1

Response: We thank the reviewer for the positive feedback and important remarks. Taking into consideration each of the reviewer’s comments, we believe we have significantly improved our manuscript by correcting typos, by clarifying text (especially Introduction section), and by adding new material. 

So crude is CD146 negative? or the original population? this is a bit unclear

Response: We have clarified the text within M&Ms section, ‘CD146+ eMSC selection’ subsection, to now read: ‘eMSC original population (designated as Crude group) was sorted based on CD146 expression to yield the CD146+ subpopulation.’

I would move ‘Isolation and validation of Crude and CD146+ eMSC EXOs’ subsection before the qPCR so that you have an order: cells, PCR, miRs etc

Response: Subsection has been transferred accordingly.

Please cite a protocol for ‘Immunopotency assay (IPA)’ subsection

Response: We have now cited in ‘Immunopotency assay (IPA)’ subsection our previous published work using this protocol an it reads: ‘Immunopotency assay was performed similarly to [10].’

ANOVA is for normal distribution. Kruskal-Wallis is for non-normal. maybe there-s a mistake here in the text?

Response: We have corrected the text in the ‘Statistical analysis’ subsection to now read: ‘One-way ANOVA was used for multiple comparisons.’

Fig1A is not cited. also in the table what exactly are those numbers? %?

Response: We have corrected the text in the Results section, ‘Crude and CD146+ eMSC characterization’ subsection and it reads: ‘On this basis, in this study eMSC were expanded until P3 and the co-expression of CD146+ and SUSD2 was evaluated, demonstrating positive expression levels of 52.97±17.85% be-tween markers. Additionally, eMSCs showed expression levels of 21.74±17.56% for CD146+/SUSD2-  and 16.87±6.09% for SUSD2+/CD146- (Figure 1A).’

cite fig1D. also the size of the text is really small like this. I would suggest to find a solution to improve this if possible

Response: We have cited within the text Figure 1D.

‘Crude and CD146+ eMSC EXOs characterization’ subsection is more of a material and meth thing than an actual result

Response: We thank the reviewer for his/her comment. This information has to be incorporated into Results section as before we start analyzing further exosomes cargo/functions, we have to 'prove' to the reader that we are working with the appropriate material. Specifically, the position statement of the International Society for Extracellular Vesicles (doi: 10.1080/20013078.2018.1535750) clearly outlines the minimum required information for studies of extracellular vesicles. Accordingly, we have followed these guidelines as presented in the Results section.

I believe a differential expression type of analysis is due here. The way you present the data it looks like you just compared a list of top expressed miRs and didn't do a statistical analysis to say which is more enriched where. Am I wrong? You could try limma in R for this analysis.

Response: We thank the reviewer for the valid comment. Indeed, differential expression analysis is ideal for this kind of comparisons, however the amount of miRNA common between the two different groups (Crude and CD146+ eMSC EXOs) was too small to perform this kind of analysis. On this basis, in future experimentation we are aiming to perform RNAseq to Crude and CD146+ eMSC EXOs and therefore yield a bigger amount of miRNA data. This will allow us to perform a more detailed statistical analysis and further identify differences between the two groups. However, even though the present study is a proof-of-concept, we are excited that Crude and CD146+ eMSC EXOs show differential miRNA cargos for our experimental design.

Figure 5B is not a good way to present the data. You can't compare the expression between crude and CD146+.

Response: We thank the reviewer for this important comment. We would like to clarify that in panel Figure 5B we are not comparing gene expression of Crude and CD146+ eMSC EXOs per se but the expression levels of THP-1 macrophages that are exposed to Crude or CD146+ eMSC EXOs treatments. Also, within the bar graph we have incorporated the 'basal' stimulated THP-1 levels without exposure to EXOs. To clarify this point we have now added in Figure 5, a figure legend and new text that reads 'Dotted line represents the stimulated THP-1 gene levels without exposure to eMSC EXOs'. This analysis and data presentation was recently used by our group for infrapatellar fat pad-derived EXOs effects on THP-1 macrophages (doi: 10.1038/s41598-022-07569-7).  

In Figure 6 state that the first images are the negative control. In Figure6C the legend is unclear. what do the 3 rows correspond to?. In Figure 6D, this has no legend. who is who?

Response: We thank the reviewer for these valid comments. We have clarified the text and added new text in Figure 6 and the figure legend to now read: 'Figure 6. eMSC EXO immunomodulatory effects on PBMC. (A) Unstimulated PBMC show unaltered morphology with and without eMSC EXO exposure (upper pan-el). Stimulation of PBMC with PMA/IO resulted in their characteristic aggregation in vitro that was not affected by exposure to eMSC EXOs (bottom panel). (B) The percentage of CFSE-labelled PMA/IO-activated PBMC proliferation (relative to unstimulated PBMC) was measured in PBMC only and after exposure to Crude and CD146+ eMSC EXOs. Exposure of PMA/IO-stimulated PBMC to two different eMSC EXO concentrations (EXOs and 2x EXOs) resulted in insignificant PBMC proliferation suppression. Every eMSC EXO donor (Crude or CD146+, n=3) co-cultured in independent experiments with four different PBMC populations obtained from unrelated donors. (C) Both Crude and CD146+ eMSC EXO exposures resulted in reduced secretion of pro-inflammatory cytokines compared to stimulated PBMC alone (marked as +). (D) 13 out of 40 molecules tested had significantly (p<0.05) lower secretion and 2 out of 40 molecules tested had significantly (p<0.05) higher secretion in eMSC EXO + stimulated PBMC cultures compared to stimulated PBMC alone (blue bar: Stim. PBMC + Crude EXOs, pink bar: Stim. PBMC + CD146+ EXOs, grey bar: Stim. PBMC). '

The STRING analysis results in Figure 6 should be presented like the genes in figure 1D for example. Or al least in a ordered table. who's negatively and who's positively enriched

Response: We thank the reviewer for his/her comments. We have created a new supplementary Figure 1, containing the ' % enrichment of PBMC secreted proteins for Biological Processes ' and the ' % enrichment of PBMC secreted proteins for KEGG and reactome pathways'. This information provides the reader a quantitative evaluation of the involvement of 13 proteins identified to have lower secretion in PBMC + EXOs in different biological processes and pathways.  

yet in the Funding section you stated that there's no funding?

Response: We have clarified within the Acknowledgments that: 'These funding sources were not involved in any step of the study design, collection, analysis, or interpretation of the data, or writing of the manuscript'.

General considerations:

The paper is quite hard to follow. The intro switches between MSC and eMSC and it's quite confusing. Figures are not cited in the text so I had to go around and search to see where the info is presented. More attention needs to be spent for these details as this sends a message of carelessness which makes one think whether the same attention was given to the experiments in general.

Response: We acknowledge that the Introduction section was confusing for the reader. On this basis, we have significantly improved the Introduction by rearranging paragraphs to provide a better transition from MSC and CD146/SUSD2 subpopulations to MSC secretomes and exosomes. We have also added Figure citations where needed to further clarify our message for the reader.  

Thp1 cell line under PMA differentiates in macrophages. Please make sure in your results that this aspect is clear. It's not easy to understand when your experiments were with Thp1 or with Thp1/Mfs.

Response: We thank the reviewer for this comment. All experimentation has been performed with differentiated THP-1 cells to macrophages namely within the text ‘PMA/IO-stimulated THP-1’. We double-checked the text to be consistent with this nomenclature.

The whole article makes comparisons between mixed (crude) population of eMSCs and CD146+ eMSC. From the graphical abstract one gets the idea that you sort the population into CD146+ and negative cells. But apparently this is not the case... Please clarify this issue. If the crude population is indeed the original one that included also CD146+ then it's indeed obvious that the crude population has similar results as the CD146+ one. Moreover, based on the result from figure 1A, it looks like most of the population is CD146+. Please comment on this. Also the graphical abstract should be changed so to show that the crude pupulation included the CD146+ one.

Response: We thank the reviewer for this comment. We have now clarified nomenclature within the figures. We have now provided to the reader data related to miRNA characterization and functional assessments with immune cells.  

I believe the graphical abstract should include the most important findings of the paper. If it gets too busy with too many elements you can separate the info and make the graphical abstract with your results (for example the M2 polarization is an important finding, the deregulated pathways, the top miRs etc) and make something similar to the original graphical abstract as a study "workflow".

Response: We thank the reviewer for this comment. We have clarified the nomenclature within the graphical abstract. We have now provided to the reader data related to miRNAs characterization and functional assessments with immune cells. 

Reviewer 2 Report

The manuscript submitted by Leñero et al. described the potential therapeutic impact of miRNAs exosomal-cargo isolated from stem/stromal cells resident in the endometrium. The topic is interesting and overall the study is well designed.

The results presented support the conclusions and the authors discussed nicely their data in respect to the current knowledge of the field in the literature.

I have just few suggestions that should be addressed:

- The flow of the Introduction section can be improved. The authors presented the rationale and previous findings supporting their work but it lacks fluency.

- It would be nice to disclose more in detail the potential impact and applications of their findings in the medicinal field (regeneration, stemness, inflammation, diseases including cancer).

- The title of the manuscript sounds a bit generic. It can be improved by highlighting the "take home message" and therapeutic implications of the research findings.

Author Response

Reviewer 2

The manuscript submitted by Leñero et al. described the potential therapeutic impact of miRNAs exosomal-cargo isolated from stem/stromal cells resident in the endometrium. The topic is interesting and overall the study is well designed. The results presented support the conclusions and the authors discussed nicely their data in respect to the current knowledge of the field in the literature.

Response: We thank the reviewer for the positive feedback about our manuscript.

I have just few suggestions that should be addressed:

- The flow of the Introduction section can be improved. The authors presented the rationale and previous findings supporting their work but it lacks fluency.

Response: We acknowledge that the Introduction section was confusing as originally presented. Accordingly, we believe we have significantly improved the by rearranging paragraphs in order to have a better transition from MSC and CD146/SUSD2 subpopulations to MSC secretomes and exosomes. We have also added Figure citations where needed within the text to further clarify our message. 

- It would be nice to disclose more in detail the potential impact and applications of their findings in the medicinal field (regeneration, stemness, inflammation, diseases including cancer).

Response: We totally agree with reviewer’s comment on the significance of correlating eMSC EXO in vitro data with their applicability to the in vivo setting. However, reviewer 1 suggested we reduce text referring to in vivo applications/diseases and due to the fact that we did not evaluate our approach in a specific animal model, we took the approach of minimizing the reporting of in vivo studies. However, we did add new text in  the first paragraph of the Discussion that reads: ‘These types of investigations could provide a rationale for further testing of the eMSC EXO as a viable therapeutic modality to manufacture cell-free products for inflammatory conditions such as osteoarthritis and diabetes where inflammation is increasingly recognized as an important component of disease progression. Specifically, macrophage infiltration and pro-inflammatory activation have been associated with synovitis/fat pad fibrosis severity in the knee joint of osteoarthritis patients [41, 42] and pancreatic islet viability/insulin secretion capacity in diabetic patients[43, 44].’

- The title of the manuscript sounds a bit generic. It can be improved by highlighting the "take home message" and therapeutic implications of the research findings.

Response: We thank the reviewer for this important point. Accordingly, we have modified the title to read: ‘CD146+ endometrial-derived mesenchymal stem/stromal cell subpopulation possess exosomal secretomes with strong immunomodulatory miRNA attributes’.

Reviewer 3 Report

This paper provides interesting  evidence about  differential  signatures of miRNA EXOs derived from crude eMSCs and CD146+  eMSC, which are strongly associated  with immunomodulatory activities on   macrophages and peripheral blood  mononuclear cells, specially that obtained from CD146+ eMSC EXOs.  This information is important to stablish various  therapeutic strategies during pathogenesis and progression of inflamatory diseases.

Author Response

Reviewer 3

This paper provides interesting  evidence about  differential  signatures of miRNA EXOs derived from crude eMSCs and CD146+  eMSC, which are strongly associated  with immunomodulatory activities on   macrophages and peripheral blood  mononuclear cells, specially that obtained from CD146+ eMSC EXOs.  This information is important to stablish various  therapeutic strategies during pathogenesis and progression of inflamatory diseases.

Response: We thank the reviewer for the positive feedback.

Round 2

Reviewer 1 Report

Thank you for taking the time to consider my comments and to answer my questions.

Please do a last grammar check after you apply all the changes to the final manuscript (some verbs are not in accordance with the subject of the sentence) and may all odds be in your favor!

Best of luck in your future projects!